# Liquid Crystals-Enabled AC Electrokinetics

**DOI:** 10.3390/mi10010045

**Published:** 2019-01-10

**Authors:** Chenhui Peng, Oleg D. Lavrentovich

**Affiliations:** 1Department of Physics and Materials Science, The University of Memphis, Memphis, TN 38152, USA; 2Department of Physics and Chemical Physics Interdisciplinary Program, Advanced Materials and Liquid Crystal Institute, Kent State University, Kent, OH 44242, USA

**Keywords:** liquid crystal, electrophoresis, electro-osmosis, alternating current (AC) electrokinetics

## Abstract

Phenomena of electrically driven fluid flows, known as electro-osmosis, and particle transport in a liquid electrolyte, known as electrophoresis, collectively form a subject of electrokinetics. Electrokinetics shows a great potential in microscopic manipulation of matter for various scientific and technological applications. Electrokinetics is usually studied for isotropic electrolytes. Recently it has been demonstrated that replacement of an isotropic electrolyte with an anisotropic, or liquid crystal (LC), electrolyte, brings about entirely new mechanisms of spatial charge formation and electrokinetic effects. This review presents the main features of liquid crystal-enabled electrokinetics (LCEK) rooted in the field-assisted separation of electric charges at deformations of the director that describes local molecular orientation of the LC. Since the electric field separates the charges and then drives the charges, the resulting electro-osmotic and electrophoretic velocities grow as the square of the applied electric field. We describe a number of related phenomena, such as alternating current (AC) LC-enabled electrophoresis of colloidal solid particles and fluid droplets in uniform and spatially-patterned LCs, swarming of colloids guided by photoactivated surface patterns, control of LCEK polarity through the material properties of the LC electrolyte, LCEK-assisted mixing at microscale, separation and sorting of small particles. LC-enabled electrokinetics brings a new dimension to our ability to manipulate dynamics of matter at small scales and holds a major promise for future technologies of microfluidics, pumping, mixing, sensing, and diagnostics.

## 1. Introduction

The microscale manipulation of fluids and colloidal particles by an electric field is an important theme in a variety of fields of knowledge, ranging from fundamental studies of active soft matter to practical applications such as information displays, medical diagnostics, biosensing, and sorting [1,2,3,4,5,6,7]. Electrokinetic phenomena are typically explored using isotropic electrolytes such as aqueous solutions. Electrokinetics relies on separation of electric charges in space. In classic linear electrophoresis, the charges are separated through dissociation of molecular groups at interfaces; the separated charges form electric double layers. A colloidal particle with such an electric double layer can move in presence of a uniform electric field, with the velocity that depends linearly on the field amplitude E, $v=\varepsilon \varepsilon_{0}\varsigma E/\eta$; here ε and η are the dielectric permittivity and viscosity of isotropic electrolyte, respectively, $\varepsilon_{0}$ is the electric constant, and ς is the zeta-potential [8]. The linear electrophoresis could be driven by a direct-current (DC) field, while a symmetric alternating current (AC) field of a zero time average produces no net displacement. It has been always desirable to find nonlinear mechanisms to transport particles and fluids, especially those with a quadratic dependence on the applied field, since in this case one can use an AC driving to power steady flows and to avoid detrimental effects such as electrode blocking and electrochemical reactions. AC electrokinetics in isotropic electrolytes has been extensively studied experimentally [9,10,11,12,13,14,15,16] and theoretically [2,17,18,19,20,21]. In 1996, Murtsovkin [17] considered nonlinear electro-osmotic flows around dielectric and ideally polarizable spheres and demonstrated that the velocities of flows around these particles grow as the square of the electric field. Later on, Bazant and Squires demonstrated that this and many other nonlinear effects can be unified under a common umbrella of the so-called induced-charge electrokinetics (ICEK) [2,18,19] which is especially effective in presence of highly polarizable (metal) surfaces or particles. In ICEK, separation of electric charges in space is caused by the electric field itself that drives the ionic currents to the electrolyte-metal interface and creates electric double layers on a macroscopic length scale, such as a diameter a of a metal sphere placed in the electrolyte; this length scale is typically much larger than the thickness of electric double layer around a dielectric particle. The field-induced potential scales as aE. The field drags the ions around the particle thus triggering electrokinetic flows. The flow velocity is proportional to $E^{2}$ since the first power of E sets up the “induced-charge” clouds and the second power drives the resultant flow [19]. If the metallic sphere is immobilized, e.g., glued to the substrate of a cell, the field-driven flow of electrolyte around the sphere is called an induced-charge electro-osmosis (ICEO). The ICEO flow trajectories around a metal sphere (Figure 1b,c) are of quadrupolar symmetry. Such a sphere shows no electrophoretic mobility [14]. However, if symmetry of the particle is broken, for example, the particle represents a dielectric-metal Janus sphere, then the ICEO flows become of a dipolar character (Figure 1e,f) [14]. Such a Janus sphere would move in a uniform DC or AC electric field, as demonstrated by Gangwal et al. [10]. 

This review discusses a new nonlinear electrokinetics that arises when an isotropic electrolyte is replaced with an anisotropic fluid, namely a nematic liquid crystal (LC). It expands the subject already presented in prior reviews [22,23,24] by incorporating the most recent advances. 

A LC exhibits a long range orientational order described by the so-called director $\hat{n}$ that specifies the average direction of local molecular orientation. All physical properties of LCs, such as electric conductivity and dielectric permittivity, are anisotropic. When the director is distorted, the electric charges can be spatially separated in presence of the electric field, because of the anisotropy of electric conductivity and dielectric permittivity. These director distortions can be created by a colloidal inclusion in the LCs or by surface patterning of molecular orientation. Similarly, to the ICEK, the elecrokinetic flows in a distorted LC show velocity proportional to the square of the field; one degree of the field separates the charges and the other degree drives the flows. The effect is called liquid crystal-enabled electrokinetics (LCEK). If the LC is uniform (in absence of colloidal inclusions) and the director distortions needed to cause propulsion of colloids are created by the colloids themselves, the LCEK effect is called liquid crystal-enabled electrophoresis (LCEP) [22]. When the distortions are produced through surface patterning at the bounding plates of the LC cell that might or might not contain colloidal particles, the effect is called a liquid crystal-enabled electro-osmosis (LCEO) [25,26]. Below, we first describe nonlinear LCEP of solid particles and fluid droplets in a LC. Unlike ICEP in isotropic electrolytes, LCEP allows one to transport particles that are spherically symmetric, since the symmetry-breaking condition is fulfilled by the surrounding director field. We then discuss LCEO in surface-patterned LCs. In the latter case, by designing specific geometries of the director, one can gain a substantial control over electro-osmotic flows, for example, create flows that would transport not only solid particles but also fluid droplets and air bubbles, produce lattices of vortices with reconfigurable polarity, reverse the polarity of electrokinetics through the change of material parameters, etc. We conclude the review with a discussion of LCEK potential in microfluidic applications such as micromixing, separation and sorting. 

## 2. Basic Properties of Colloids in LC 

The simplest type of a LC, the so-called nematic, which we deal with in this review, exhibits a long-range orientational order but no positional order [27]. The director is non-polar, $\hat{n}=-\hat{n}$. All physical properties of LCs are anisotropic. If a nematic experiences a director reorientation over a length scale *L*, the elastic energy cost can be estimated as *KL*, where *K* is the Frank elastic constant with an order of magnitude 10 pN. If the boundary sets the director to align along some preferred axis, called the “easy axis” ${\hat{n}}_{0}$, any deviation of the actual $\hat{n}$ from ${\hat{n}}_{0}$ will require some work [27]. The measure of this work is called the surface anchoring strength, *W*. The ratio *λ* = *K*/*W* has a dimension of length, and is called the de Gennes-Kleman length. The typical anchoring strength is *W* ~ (10^−6^–10^−3^) J/m^2^ [28], which yields *λ* = 0.1–10 μm, which is roughly the scale that corresponds to colloidal particles. As a result, confined LC droplets and colloidal inclusions show an interesting change in behavior when their size is much smaller than *λ* and much larger than *λ* [29,30]. For droplets of LC and inclusions in LC of size *a* much smaller than *λ*, the cost of violating surface anchoring conditions is much smaller than the cost of elastic distortions, $Ka/Wa^{2}\sim \lambda /a>>1$, thus the director field within the droplets or outside the inclusions is practically uniform and deviates significantly from the “easy axis”. If a>>λ, the situation is reverse, as the elastic energy of elastic distortions ∼Ka becomes smaller than $∼Wa^{2}$, $Ka/Wa^{2}\sim \lambda /a<<1$, thus the director field is strongly distorted in order to satisfy the surface orientation as close as possible. 

Concrete geometry of director distortions around large spheres of a radius a>λ depends on the type of surface anchoring. If the surface anchoring requires the director to be perpendicular to the surface, such a sphere, when placed in an otherwise uniform nematic, would produce a distorted director with a point defect, the so-called hyperbolic hedgehog, which serves to match the local radial director and the uniform far-field director [31], Figure 2a. The structure is of a dipolar symmetry. The hedgehog has an equal probability to be of the left or right side of the sphere. The structure can be characterized by introducing an elastic dipole vector p directed from the hedgehog toward the sphere. When the sample thickness is comparable to the diameter 2 *a*, or when there is an electric or magnetic field applied to align the director, the hyperbolic hedgehog transforms into a disclination ring around the sphere [32,33] (Figure 2b). A sphere with tangential anchoring in the uniform nematic produces a configuration of quadrupolar symmetry with two surface point defects-boojums at the poles (Figure 2c). As will be described below, colloids with dipolar and quadrupolar symmetry show very different dynamics in an external electric filed. 

## 3. LC-Enabled Electrophoresis

In presence of a uniform electric field, dielectric (glass) and metal (gold) spheres with a dipolar structure shown in Figure 2a become electrophoretically active in a LC, moving with a speed that grows with the square of the electric field, $v\propto E^{2}$ (Figure 3) [34]. If the LC is melted into an isotropic phase by heating, this effect disappears, as the glass spheres show only a linear electrophoresis, while the gold spheres do not move at all [34].

The quadratic relationship in this LCEP effect is similar to that in ICEP, but the underlying mechanisms and features are different.
(i)In ICEP, only asymmetric particles can move, such as Janus spheres, while LCEP allows one to move perfect spheres; the reason is that it is the LC electrolyte environment that breaks the fore-aft symmetry [34]. (ii)LCEP is effective in carrying both dielectric and metallic particles [34] as the charge separation occurs at the director field deformations that are determined by the type of surface anchoring but are less dependent on the material of the particle. (iii)The elastic dipole p of a normally anchored sphere in a LC is usually parallel to the overall director field [34,35] and cannot reverse its polarity; as a result, the particle’s trajectory is linear (parallel to the average director around it) and polar, as dictated by the direction of p (Figure 3). In ICEP, the structural dipole of Janus spheres can adopt any orientation perpendicular to the electric field and its trajectory is hard to control [10]. (iv)As will be shown below, the polarity of LCEP trajectories can be controlled by changing the anisotropy sign of conductivity and dielectric permittivity [36].

The feature (ii) in the list above has been demonstrated in spectacular experiments by Hernàndez-Navarro et al. [37] on LCEP transport of water droplets dispersed in a LC host. The surface anchoring at these droplets was set to be perpendicular by adding surfactants. Under the action of an AC electric field, these droplets move parallel to the far-field director ${\hat{n}}_{\mathrm{FF}}$ (Figure 4a) with a speed $v\propto E^{2}$ (Figure 4b). 

LCEP shows interesting features in terms of collective behavior of colloidal particles. In particular, Sasaki et al. [38] demonstrated that single particles and multi-particle self-assembled chains, including chains with an embedded micro-cargo, can be transported through the LC medium by combining LCEP and electrohydrodynamic convection. Another remarkable collective effect of “swarming” was demonstrated by Sagués’ group [39,40,41]. The particles in the experiments were of a pear-like shape. This shape assures that the symmetry of the director field is dipolar even if the surface anchoring of the director is quadrupolar. The particles are thus LCEP-active. The LC cell is bounded by a substrate with a photosensitive substrate that can be tuned to yield a planar anchoring under an ultraviolet (UV) (365 nm) irradiation and a perpendicular (called also homeotropic) anchoring under visible light (455 nm) irradiation. The initial state of the LC cell is homeotropic and the LC is of a negative dielectric anisotropy. Illuminating the cell with a UV light realigns the LC molecules parallel to the photosensitive plate. The director forms a radial structure seen as a Maltese-cross texture under a polarizing microscope (Figure 5a). When the electric field is applied, it reinforces the tangential alignment and also drives the pear-like particles towards the center of the pattern, forming an aster-like assembly (Figure 5b,c). This radial structure can be transformed into a spiral one, by shining a blue light at a small central region to enforce a homeotropic alignment. When the electric field is applied, the resulting structure is of a spiral type, combining splay and bend deformations (bend elastic constant is smaller than the splay elastic constant in the material explored [39]) (Figure 5d). The particles driven by LCEP to the center region assemble into a rotating vortex (Figure 5e,f). Even more intriguing, the aster and spiral clusters of colloids can be moved anywhere in the plane of the LC cell, by changing the location of the UV spot (Figure 5g [39,41]). All these properties allow one to design and address dynamically reconfigurable 2D lattices of clusters [39]. 

Another interesting example of collective dynamics powered by LCEP is shown in Figure 6 for a phase-separated dispersion of silicone oil droplets in a nematic LC. When a nematic LC material E7 is doped with a small amount (1 wt%) of silicone oil (from Sigma-Aldrich, St. Louis, MO, USA) and heated above its melting temperature, to 75 °C for 2 min, the oil and LC form a homogeneous mixture. When the sample is quenched to the room temperature, a biphasic region appears, with the oil droplets nucleating within the LC environment and growing till some critical radius a~λ. Further growth of the droplets is hindered, since the director field around them becomes distorted and produces hyperbolic hedgehogs; these distortions around neighboring oil droplets repel each other and prevent coalescence [42]. If the LC medium is uniformly aligned, one observes periodic arrays of linear self-assembled chains of oil droplets [42,43], Figure 6. When an AC electric field (*E* = 30 mV/μm, frequency 5 Hz) is applied to such a system, LCEP moves the chains to one side of the chamber, as specified by the polarity of the structural dipoles (Figure 6a–c). 

To clarify how the electric field separates charges in the LC medium around spherical colloids and causes LCEP, Lazo et al. [25] performed experiments on spheres of various surface anchoring types that were glued to the glass substrates. In the presence of an electric field, these spheres induce LCEO flows. Since the spheres are immobilized, the flows are easy to explore by adding fluorescent tracers to the LC [25]. Consider a sphere with perpendicular anchoring on the surface adopting the “Saturn ring” configuration in the LC with director $\hat{n}$ (Figure 7a). An electric field E applied along *x*-axis moves the charges along this axis (positive charges follow the positive direction of the *x*-axis). However, the charges also shift along the *y*-axis. The shift along the *y*-axis is caused by anisotropy of conductivity, which is usually positive, $\Delta \sigma =\sigma_{\left|\right|}-\sigma_{⊥}>0$. Here and in what follows the subscripts refer to the principal components measured along the direction parallel and perpendicular to the director. This anisotropy guides the charges along the director field. For the field polarity in Figure 7a, director deformations that converge at the sphere because of perpendicular anchoring, guide positive charges towards the left hand side of the sphere, while the negative charges are gathered on the right hand side. This induced charge density is proportional to the applied field, ρ∝E. The separated charges experience a Coulomb force of density f∝ρE, which triggers electro-osmotic flows around the sphere (Figure 7a). When the field polarity is reversed, the polarity of ion clouds is also reversed, but the driving force $f\propto \rho E\propto E^{2}$ preserves its sign, which yields the flows polarity-insensitive with velocities that grow as $E^{2}$. As a result, an AC electric field of a frequency that is not too high (to allow the charges a sufficient time to separate in space), powers steady electro-osmotic flows around the sphere (Figure 7b) [25]. The flows in Figure 7b are of quadrupolar symmetry, following the quadrupolar symmetry of the director in Figure 7a. Because of this symmetry, such a sphere, if freely suspended in an LC, would not show a directional motion. However, if the sphere with perpendicular anchoring adopts a dipolar director configuration (Figure 7c) the symmetry is broken, and a dipolar flow pattern is induced (Figure 7d). The LCEO pumps the nematic fluid from right side to the left in Figure 7d. If the particle in Figure 7d is free to move, it would move from left to right, with a velocity $\propto E^{2}$, which explains the mechanism of AC driving LCEP of dipolar particles [34] shown in Figure 3. 

To estimate the typical velocities involved in LCEO and LCEP, let us calculate, following Lazo et al. [25], the density ρ of charges created as a result of conductivity anisotropy in a distorted LC and establish how it depends on the typical scale a of director distortions, E, and Δσ. For simplicity, consider two dimensional geometry and assume that the nematic dielectric anisotropy is small, $\Delta \varepsilon <<\bar{\varepsilon }$, where $\bar{\varepsilon }=\left(\varepsilon_{\left|\right|}+\varepsilon_{⊥}\right)/2$, and that the director distortions are weak, $\hat{n}=\left(1,\varphi \right)$, where φ=φ(x,y)<<1 is a small tilt angle that the director makes with the *x*-axis. The field, applied along the *x*-axis, creates ionic currents $J_{i}=\sigma_{ij}E_{j}$, where $\sigma_{ij}=\sigma_{⊥}\delta_{ij}+\Delta \sigma n_{i}n_{j}$ is the conductivity tensor; *i* and *j* stand for *x* and *y*. For small φ’s, the current components are $J_{x}=\sigma_{\left|\right|}E_{x}+\Delta \sigma \varphi E_{y}$ and $J_{y}=\sigma_{⊥}E_{y}+\Delta \sigma \varphi E_{x}$; note the non-zero field component $E_{y}$ induced by separation of charges. Using the charge conservation law, divJ=0 and Poisson’s equation divD=ρ (where D is the electric displacement) one obtains the charge density ρ(x,y) caused by conductivity anisotropy Δσ and director gradients |∂φ/∂y|~1/a: (1)$$\rho \left(x,y\right)=\left(-\frac{\Delta \sigma }{\bar{\sigma }}+\frac{\Delta \varepsilon }{\bar{\varepsilon }}\right)\varepsilon_{0}\bar{\varepsilon }E_{x}\frac{\partial \varphi }{\partial y}$$
here $\bar{\sigma }=\left(\sigma_{\left|\right|}+\sigma_{⊥}\right)/2$ is the average conductivity. The space charge experiences a bulk force of density $f\propto \rho E_{x}$ that drives the LCEO flows. If the particle under consideration is free to move, then its LCEP velocity can be estimated by balancing the Coulomb force f and viscous resistance $\eta u/a^{2}$:(2)$$v=\pm \beta \frac{\varepsilon_{0}\bar{\varepsilon }}{\eta }\left(\frac{\Delta \varepsilon }{\bar{\varepsilon }}-\frac{\Delta \sigma }{\bar{\sigma }}\right)aE_{x}^{2}$$

Here the numerical coefficient β on the order of 1 is introduced to account for the approximations such as using 1/a as a measure of director gradients and replacing anisotropic viscosity of the LC with its average value η; the sign “+” or “−“ depends on the polarity of director gradients such as ∂φ/∂y. For an extended theoretical treatment of LCEK involving colloidal particles, see Refs. [36,44,45,46,47].

Equation (2) for the LCEP velocity suggests that the efficiency and even polarity of LCEP transport can be controlled by tuning the material parameters, namely, the value of factor $\left(\frac{\Delta \varepsilon }{\bar{\varepsilon }}-\frac{\Delta \sigma }{\bar{\sigma }}\right)$. The latter varies broadly and even changes its sign in mixtures of different LCs and in some cases when the temperature of the LC is changed [36]. As a result, one can reverse the polarity of LCEP by composition (Figure 8a) and by temperature (Figure 8b). 

## 4. LC-Enabled Electrokinetics in Patterned Cells

In LCEK discussed so far, the director distortions are caused by colloids with certain surface anchoring. The approach is limited, as the colloid needs to be sufficiently large and even then it can produce only a finite set of director patterns; some of these are not suitable for transport or pumping. For example, a sphere with tangential alignment does not show any electrophoretic activity since the director field around it is of quadrupolar symmetry [25,34]. However, the mechanisms of LC-enabled AC electrokinetics is not limited by colloids of certain surface anchoring, since the director distortions in LCs can be imposed by various means, in particular, by patterned surface anchoring at the bounding plates [48,49,50,51,52].

A convenient and versatile approach to pattern surface alignment is by photoalignment [53,54,55,56] based on plasmonic masks with nanoslits [51]. When such a mask is illuminated with non-polarized light, the slits transmit a polarized optical field; the direction of polarization is perpendicular to the long axis of the nanoslit. The pattern of nanoslits is predesigned in accordance with the desired LC alignment. The transmitted optical field with the spatially varying linear polarization is projected onto a layer of photosensitive material such as azodye Brilliant Yellow, deposited on a glass substrate. The dye molecules align their long axes perpendicularly to the local polarization of light to avoid trans–cis isomerization. The resulting patterned layer of dye is used to align the LC molecules. The director pattern of the LC thus reproduces the predesigned pattern of nanoslits. Usually, top and bottom plates of a LC cells are photoirradiated at the same time, so that the director field does not change along the normal to the bounding plates, although one can also use differently treated substrates. 

Peng et al. demonstrated how the prepatterned director fields create LCEO flows in the LC cell [26]. One of the examples is illustrated in Figure 9. The director is tangential at the bounding plates and periodically modulated along the *y*-axis, with a period 2*l*: (3)$$\hat{n}=\left(n_{x},\;n_{y},0\right)=\left(\left|\cos\frac{\pi y}{l}\right|,\;\sin\frac{\pi y}{l},0\right)$$

Consider the behavior of ions in such a cell. In absence of the electric field, the ions are distributed homogeneously in the sample. Once the electric field $E=\left(E_{x},\;0\right)$ is applied, the ions move along the *x*-axis, but they also shift along the vertical *y*-axis, because they prefer to move parallel to the director rather than perpendicularly to it, Δσ>0 (Figure 9a). The field separates the charges in space, forming alternate lanes of opposite polarity, Figure 9b. The charge density $\rho \left(y\right)=\varepsilon_{0}\bar{\varepsilon }\frac{\partial E_{y}}{\partial y}$ is determined by the transverse electric field, $E_{y}=-\Delta \sigma \frac{\cos\alpha \sin\alpha }{\sigma_{∥}\cos^{2}\alpha +\sigma_{⊥}\sin^{2}\alpha }E_{x}$, where α(y) is the angle between the local director $\hat{n}$ and the *x*-axis [26]. The space charge experiences a Coulomb bulk force of density f=ρE, Figure 9c, that drives electro-osmotic flows. Reversing polarity of E reverses polarity of ρ, but the product f=ρE does not change (Figure 9d). As a result, the LCEO flows in a patterned director field can be driven by an alternating current (AC) field (Figure 9e). Dielectric anisotropy supplements the mechanism of LCEO with an additional source of space charge [26,46,47].

The LCEO flows can be used to transport particles of any kind, such as tangential anchored polystyrene spheres dispersed in the LC (Figure 9f–h), which does not show any electrophoretic activity under AC electric field. It has also been used to drive droplets of fluids such as water and even gas bubbles [26]. LCEO directed by surface patterning does not impose any limitations on the properties of the “cargo” (such as presence of electric double layers, polarizability, or ability to distort the LC). 

The LCEO effect is similar to the broad class of hydrodynamic phenomena often called the Carr–Helfrich effect [57,58]. The difference is that in the Carr–Helfrich effect, electrohydrodynamic flows occur in an initially uniform LC, perturbing the director. In LCEO, the director distortions are predesign to separate the charges.

In classic linear electrokinetics, the fluid velocity v is proportional to the electric field and the resulting flows are irrotational, ∇×v=0. For practical applications such as mixing, flows with vortices are desirable [59]. Vortices are very easy to produce by LCEO in patterned LC cells, by using localized surface patterns, for example, with topological defects. The topological defects offer yet another degree of freedom in manipulating colloids as they can be used for entrapment and release [60,61,62]. Director patterns with pairs of disclinations of strength (*m* = ½, −½) and triplets such as (*m* = −½, 1, −½) and (*m* =½, −1, ½) (Figure 10a–c) produce LCEO vortices once an AC electric field is applied (Figure 10d–f). The defect pair in Figure 10a acts as a pump, pumping the LC from ½ defect towards the −½ defect. The triplet (−½, 1, −½) in Figure 10b produces a flow of the “pusher” type, with the fluid moving from the central +1 defect towards the two −½ disclinations at the periphery (Figure 10e) while the triplet (½, −1, ½) (Figure 10c) produces a “puller” flows (Figure 10f). The reason is that the LCEO flow velocities are linearly dependent on the director gradients. 

Surface patterning offers a broad range of possibilities in the design of flows. For example, two-dimensional array of LCEO vortices with clockwise and counter-clockwise circulation can be produced by a two-dimensional lattice shown in Figure 11 [26]. Importantly, the polarity of each and every vortex can be reversed by a simple reorientation of the electric field, from $E=\left(E_{x},E_{y},E_{z}\right)=\left(E,0,0\right)$ (Figure 11c) to E=(0,E, 0) (Figure 11d). Note also that photoalignment of LCs can in principle be repeatedly written and re-written [63,64], which provides another degree of freedom.

From the fundamental point of view, LCEK systems in general and LCEO in particular represents a distinct type of an active matter in which the energy input that drives the system out of equilibrium provided by spatially localized distortions of molecular orientation, either at the colloidal particles [34,35] or at patterned substrates [26]. Recently, Conklin, Viñals, and Valls [65] demonstrated that LCEO flows in patterned director fields with topological defects can model the flows experimentally observed in active matter [55]. In particular, the Coulomb body force acting on ions in a prepatterned director field, is of the type
(4)$$f\propto \xi \left(\hat{n}\mathrm{div}\hat{n}-\hat{n}\times \mathrm{curl}\hat{n}\right)$$
where the coefficient $\xi =\left(\frac{\Delta \varepsilon }{\varepsilon_{⊥}}-\frac{\Delta \sigma }{\sigma_{⊥}}\right)$ determined by the conductivity and permittivity anisotropies, plays the role of “activity” in the similar expression for an active force in active matter [66,67] and the activation force in thermally responsive elastomers [68]. 

## 5. Applications of LC-Enabled AC Electrokinetics

In the previous sections, we described transport of colloids by LCEP and creation of LCEO flows in patterned LCs. Both LCEP and LCEO can be used in applications such as controlled microreactions [37,40], accelerated micro-mixing [26], and sensing and sorting of particles that are identical in bulk properties but differ in subtle surface features [69,70]. 

Figure 12a depicts a microreaction scheme in which water droplets with perpendicular surfactant-enforced anchoring are driven by LCEP in a uniform nematic LC [37]. Two water droplets are loaded with specific reactants and move towards each other. When they coalesce into a single droplet, a chemical microreaction is triggered (Figure 12a) [37]. Similar microreaction effect can be designed through photo-patterned LCEO flows. As already indicated, a defect pair (½, −½), serves as an LCEO pump that drives the LC from the periphery of the chamber towards the core of the ½ disclination (Figure 10a,d). When water droplets are dispersed in such a LC chamber, they are driven towards the core of the ½ disclination and coalesce there (Figure 12b). Thus, the core of the ½ disclination serves as a predesigned and fixed location of a microreaction. Note that the water droplets are carried by the LCEO flows in prepatterned nematic environment even when their surface anchoring is tangential; in this case, there is no need to add surfactants to the system in order to impose perpendicular anchoring.

LCEO flows in photo-patterned director fields can facilitate microscale mixing (Figure 13). A predesigned pattern that triggers vortices of LCEO flows is placed after a Y-junction in a microfluidic channel [71]. The top inlet is injected with a LC containing 200 nm fluorescent tracers and the bottom inlet is injected with a pure LC (Figure 13a,b). If there is no electric filed, mixing is achieved only by slow diffusion. When the AC electric field is applied, a flow pattern with vortices (Figure 13c) accelerates mixing. Figure 13d compares the efficiency of mixing through the standard deviation δ of the fluorescent intensity, normalized [59,72] in such a way that the unmixed state corresponds to δ = 1, while completely mixed state shows δ = 0. The process is much faster when assisted by LCEO flows (Figure 13d). As compared to other micro-mixers [73], the LCEK mixers do not require any moving mechanical parts, pressure gradients, nor complicated patterns of electrodes and ridges obstructing the flow. Since the LCEK effect is nonlinear and since the charges are separated by anisotropic properties of the LC electrolyte, much lower voltages are needed and the electric field can be uniform, which significantly simplifies the design of chambers. Note, however, that the potential applications are limited by the thermotropic (usually hydrophobic) nature of LCs used in current LCEK; development of lyotropic hydrophilic LCs for LCEK would significantly expand the range of potential applications.

Anisotropic surface properties of LCs make them sensitive to chemical and physical properties of the adjacent media [74]. Various chemical and biological sensors have been demonstrated based on these anisotropic interfacial properties [75,76,77,78,79]. LCEK allows one to expand these approaches, as it offers a unique possibility to sort and separate microparticles that are identical to each other in terms of bulk properties (size, density, optical appearance), but differ only in surface functionalization [70], as briefly explained below. 

The director field used for the separation of particles by their surface properties is of the type
(5)$$\hat{n}=\left(n_{x},\;n_{y},0\right)=\left(\cos\frac{\pi y}{l},\;\sin\frac{\pi y}{l},0\right)$$
which forms an alternating regions of deformation of splay and bend along the *y*-axis (Figure 14a). Consider two types of spheres in such a pattern, one with a tangential anchoring and one with a perpendicular anchoring, in such a patterned nematic. As demonstrated in Refs. [69,80], the tangentially anchored spheres accumulate in the bend regions, while the perpendicularly anchored spheres move towards the regions of splay, as shown in Figure 14a. The reason is that these placements reduce the overall energy of elastic deformations of the director. If there is no electric field, the spheres of different surface anchoring thus would be separated along the *y*-axis. By applying a horizontal electric field that triggers LCEO, they can be also moved toward the opposite ends of the chamber along the *x*-axis. In presence of the electric field, the periodic director in Equation (5) serves to separate the charges; for the direction of field shown in Figure 14a, the positive charges accumulate in the regions of bend, the negative charges accumulate in the regions of splay. Reversal of the field polarity reverses the polarity of charges, but the Coulomb force preserves its sign, as in the other examples of LCEK considered above. As a result, the LCEO flows in the splay regions carry the spheres with the perpendicular anchoring to the right and the flows in the bend regions carry the spheres with tangential anchoring to the left. Note that the effective viscosity along the bend and splay regions is somewhat different, because the flow is parallel to the local director in the splay regions and perpendicular to it in the bend region. The pre-designed pattern in Figure 14a thus serves dual functions. First, it separated the colloids along the *y*-axis thanks to the elasticity of the nematic and second, it separates the colloids along the *x*-axis by the LCEO flows. 

An experimental implementation of the sorting and separation principle is illustrated in Figure 14b–d for separation of glycerol droplets with tangential anchoring [81] and glycerol droplets with surfactant sodium dodecyl sulfate (SDS) that sets perpendicular anchoring. The droplets with SDS migrate to the splay region while pure glycerol droplets to the bend region. When an AC electric field is applied, the LCEO flows carry the glycerol droplets with SDS to the left side while the pure glycerol droplets to the right. Sensing and sorting can also be applied to other materials, as long as they set different surface anchoring at the particle-LC interface [70]. For instance, 1,2-Dilauroyl-sn-glycero-3-phosphorylcholine (DLPC), a phospholipid found in biological membranes, can also be sensed and sorted by this method. In Figure 14e–g, the glycerol droplets with perpendicular anchoring caused by DLPC are transported to the left, while the pure glycerol droplets in the bend regions move to the right [70]. 

## 6. Conclusions

We discussed the broad theme of liquid crystal-enabled electrokinetics (LCEK). Usually, the carrier medium in electrokinetic phenomena is an isotropic electrolyte. Replacing the isotropic electrolyte with a liquid crystal brings a new mechanism of spatial charge generation that is based on anisotropy of conductivity and permittivity and presence of the director deformations. Under an applied electric field, these director distortions serve as regions in which the field-induced spatial charges accumulate. Once the charges are separated in space, the Coulomb force triggers either electrophoresis or electro-osmosis, depending whether the non-liquid crystal component is free to move or is immobilized. As a result, the electrophoretic and electro-osmotic velocities in LCEK are proportional to the square of the electric field: one degree of the field separates the charges in space, the second drives their flows. This mechanism lifts many limitations on the properties of particles that can transported by LCEK, since the features such as polarizability, zeta-potential, and other material properties of the particles (including whether they are solid, fluid or even gas) are of a little importance. The quadratic dependence on the electric field amplitude means that LCEK effect can be driven by an AC field with a single harmonic and zero mean value. AC driving is advantageous over the DC driving as it allows one to produce steady flows and mitigate detrimental electrode effects such as voltage blocking and electrochemical reactions. 

The mechanism of LCEK requires the director field to be distorted. Such a distortion can be achieved by colloidal particles dispersed in the liquid crystal, through the so-called surface anchoring of the director. Not all surface anchoring types can impart electrophoretic activity, however. Only those director distortions of the liquid crystal that break fore-aft symmetry can result in electrophoretic propulsion; for example, spherical colloids with tangential surface anchoring produce a director field that is quadrupolar and thus they are not electrophoretically active. 

A more general approach to induce LCEK is through surface patterning of the director at the bounding plates of the liquid crystal cells. When the electric field is applied to such a pre-patterned liquid crystal, it induces spatial charges that reflect the geometry of director and then drives LCEO flows of these charges. Any object placed in such a LC medium will be driven, regardless of its material properties, including its surface properties; in a prepatterned nematic, even a tangentially anchored sphere will be carried by the LCEO flows. 

The LCEO and LCEP trajectories are determined by the predesigned pattern of molecular orientation, by the direction of the applied electric field, and by the anisotropy of the material. In particular, changing the sign of material anisotropy reverses the direction of LCEP [36], while change in the direction of the electric field reverses the polarity of LCEO vortices [26]. Vortex generation [25,26] is also an important feature of LCEK, suitable for micromixing and rooted in the fact that the velocities of flows are growing with $E^{2}$. In LCEO, the cross-sections of the patterned LC microfluidic chambers are not obstructed by any barriers (such as ridges, electrode posts or colloidal particles, needed in other electrokinetic devices). This feature combines efficiency of flows with simplicity of design. 

The features of AC-driven LCEK outlined in this review suggest strongly that the phenomenon might find applications in lab-on-the-chip and microfluidic devices, in electric manipulation of soft and hard matter, pumping, mixing, sensing and diagnostics. Absence of pumps that generate pressure gradients is another obvious advantage of LCEK.

There are a number of issues related to LCEK that are worth exploring in the near future. As already indicated, LCEK represents a distinct type of an active matter in which the energy input that drives the system out of equilibrium provided by spatially localized distortions of the director. For some sets of parameters, as discussed by Conklin, Viñals and Valls [65], LCEK flows can be used to model active matter, which is a welcome development since many experiments on active matter involving living elements are hard to control. Another avenue of research could explore nonlinearities of the order higher than 2. In the theoretical treatments performed so far, the LCEK velocities were shown to be proportional to $E^{2}$, see Equation (2). The model leading to Equation (2) assumes that the applied field does not modify the director field substantially. However, if the coupling of the director to the electric field become strong, one would expect higher order of nonlinearity to appear in the field dependencies of velocities. Finally, practical applicability of LCEK would be expanded if one could develop water-based lyotropic liquid crystals with LCEK features similar to those in the thermotropic hydrophobic liquid crystals considered so far. 

## Figures and Tables

**Figure 1 micromachines-10-00045-f001:**
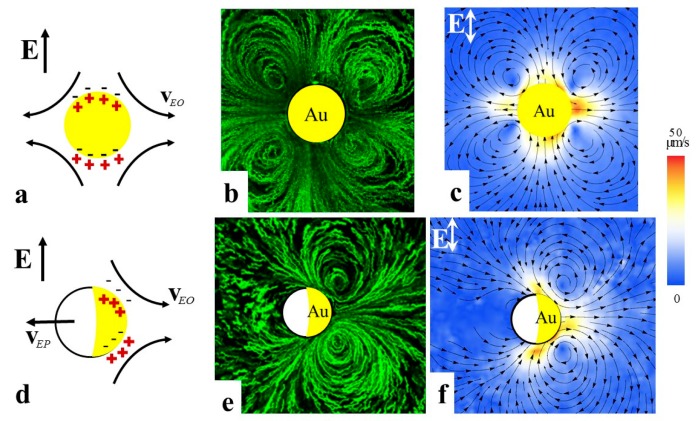
Induced-charge electro-osmosis (ICEO) around an immobilized gold sphere (**a**–**c**) and a Janus glass-gold sphere (**d**–**f**) as established by Peng et al. [14]. (**a**,**d**) Scheme of charge separation in presence of the electric field; (**b**,**e**) Experimental flow patterns driven by an AC field and visualized by fluorescent tracers; (**c**,**f**) Velocity fields; note quadrupolar symmetry of flows around the metal sphere in (**b**,**c**) and broken left-right symmetry of flows around the immobilized Janus sphere which acts as a pump; the flows are stronger on the Au (right) side than on the glass (left) side of the sphere in (**e**,**f**) [14]. Adapted with permission from [14].

**Figure 2 micromachines-10-00045-f002:**
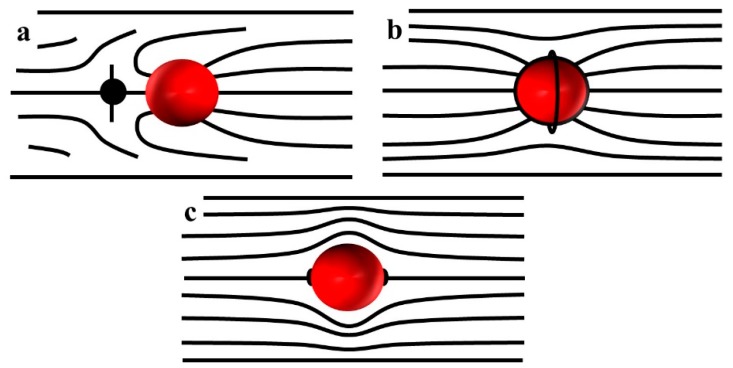
Director configurations around colloidal particles in a uniform nematic. (**a**) Dipolar structure with a hyperbolic point defect-hedgehog on the left hand side of a sphere with normal surface anchoring; (**b**) Quadrupolar structure with an equatorial director field around a sphere with normal anchoring in a shallow nematic cell; (**c**) Quadrupolar deformations around a tangentially anchored sphere with two point defects-boojums at the poles.

**Figure 3 micromachines-10-00045-f003:**
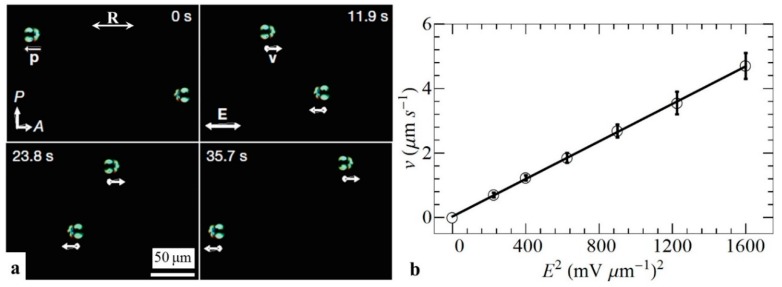
Nonlinear liquid crystal-enabled electrophoresis (LCEP) of two glass spheres powered by an alternating current (AC) electric field demonstrated by Lavrentovich et al. [34]. (**a**) Time sequence of polarizing microscope textures of two colloidal spheres, moving in opposite directions defined by the orientation of the structural dipole **p**; **R** represents the rubbing direction which sets the uniform alignment of liquid crystal (LC); (**b**) Quadratic dependence of spheres velocity on the amplitude of the AC electric field. Adapted with permission from Ref. [34].

**Figure 4 micromachines-10-00045-f004:**
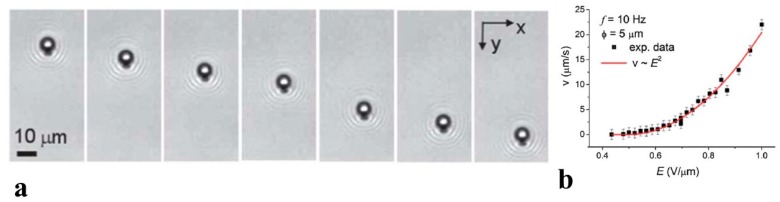
LCEP of water droplets driven by an AC electric field, as demonstrated by Hernàndez-Navarro et al. [37]. (**a**) Time sequence of optical microscopy of textures of a droplet with dipolar director field and a hyperbolic hedgehog; (**b**) Quadratic dependence of droplet velocity on the amplitude of the AC electric field. Adapted with permission from [37].

**Figure 5 micromachines-10-00045-f005:**
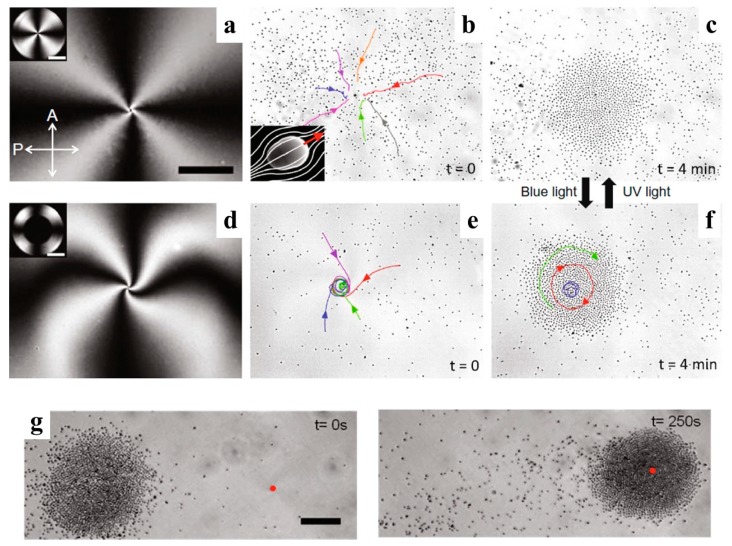
Reconfigurable swarming of anisometric particles driven by LCEP observed by Hernàndez-Navarro et al. [39]. (**a**) Image of photoactivated texture of LC with radial configuration; (**b**,**c**) Formation of colloidal aster by LCEP; (**d**) Image of photoactivated texture of LC with spiral configuration; (**e**,**f**) Assembly of a rotating cluster by LCEP; (**g**) LCEP swarming of particles by in situ reconfiguration of LC director field. Adapted with permission from [39].

**Figure 6 micromachines-10-00045-f006:**
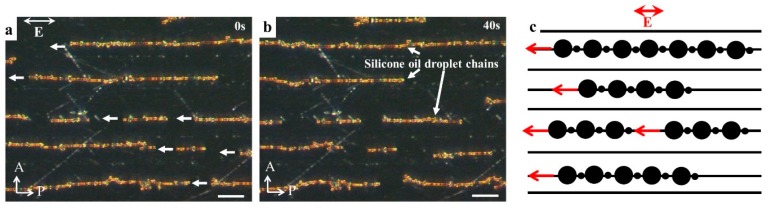
LCEP of ordered silicone oil droplet assemblies in LC. (**a**,**b**) Time sequence of the polarizing microscopy textures of silicone oil droplet chains formed by phase separation from the LC; (**c**) Scheme of LCEP transport of droplet chains. Each droplet is accompanied by a hyperbolic hedgehog shown as a small black disk. Scale bar is 50 μm. P and A represent polarizer and analyzer.

**Figure 7 micromachines-10-00045-f007:**
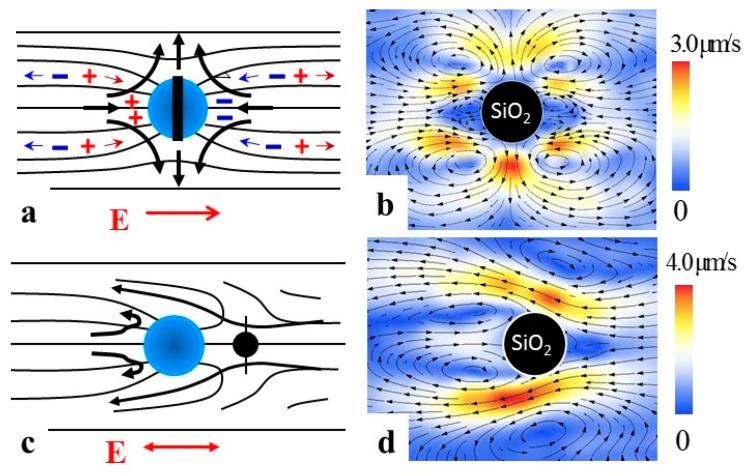
Liquid crystal-enabled electro-osmosis (LCEO) around a sphere with perpendicular anchoring and quadrupolar (**a**,**b**) or dipolar (**c**,**d**) director distortions; left column shows LCEO flows (thick arrows) and the director (thin lines). Right column shows experimental flow patterns around immobilized glass spheres of diameter 50 μm; AC field, 5 Hz. Adapted with permission from Lazo et al. [25].

**Figure 8 micromachines-10-00045-f008:**
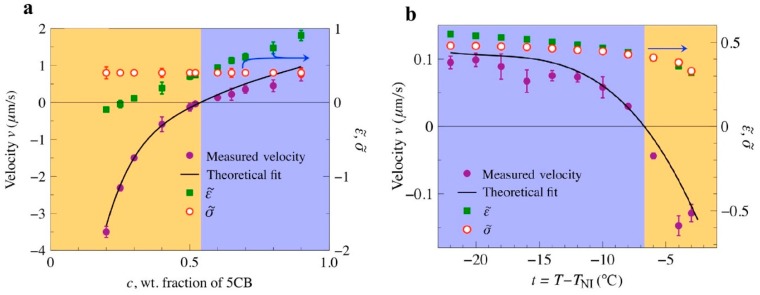
Sign reversal of LCEP velocity by changing anisotropic permittivity of LC 5CB-HNG 715600-100 mixture, demonstrated by Paladugu et al. [36]. (**a**) Dependence of LCEP velocity, anisotropic permittivity and conductivity on the concentration of LC one component, 5CB; $\tilde{\varepsilon }=\frac{\varepsilon_{∥}}{\varepsilon_{⊥}}-1$ and $\tilde{\sigma }=\frac{\sigma_{∥}}{\sigma_{⊥}}-1$; (**b**) Dependence of LCEP velocity, anisotropic permittivity and conductivity on temperature. Adapted with permission from [36].

**Figure 9 micromachines-10-00045-f009:**
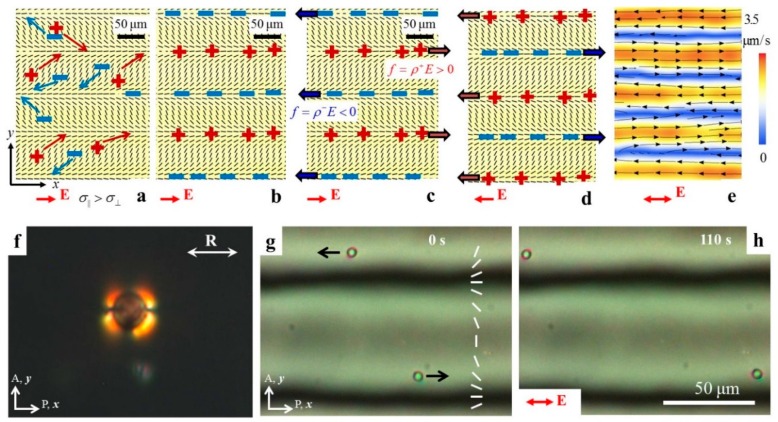
LCEO in an LC electrolyte with a predesigned director. Ticks represent local director visualized by LC PolScope: (**a**) An electric field E drives “+” and “−“ electric charges along the *x*-axis but because of the anisotropic conductivity, the charges shift also along the *y*-axis, (**b**) Creating separated lanes of space charges; (**c**) The electric field moves the charged clouds along the *x*-axis; (**d**) Reversal of the field changes polarity of charges but does not change the Coulomb force; (**e**) An AC field of frequency 5 Hz creates lanes of LCEO flows of alternating direction; (**f**) Polarizing microscopy texture of a polystyrene sphere with tangential anchoring and bipolar director field; (**g,h**) Time sequence of AC-driven LCEO flows carrying polystyrene spheres with tangential surface anchoring. Parts (**a**–**e**) and (**g**,**h**) are adapted with permission from Peng et al. [26].

**Figure 10 micromachines-10-00045-f010:**
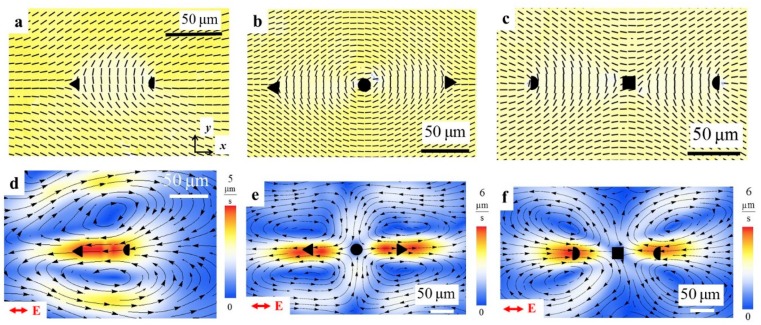
LCEO flows with pairs and triplets of topological defects demonstrated by Peng et al. [26]. (**a**) PolScope texture of a nematic cell with disclination pair −½ (core marked by a triangle) and ½ (core marked by a semicircle); (**b**) Three disclinations (−½, 1, −½) with +1 defect core marked by a circle; (**c**) Three disclinations (½, −1, ½) with –1 defect core marked by a square; (**d**–**f**) Corresponding LCEO flow patterns caused by an AC field in the patterned director fields shown in (**a**–**c**) respectively. Adapted with permission from [26].

**Figure 11 micromachines-10-00045-f011:**
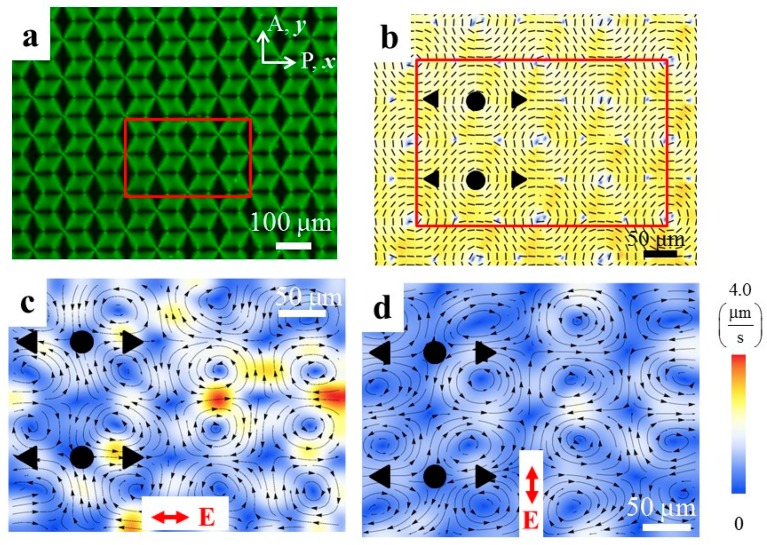
LCEO with surface patterned two-dimensional lattice of topological defects [26]. (**a**) Polarizing microscopy texture of a periodic array of disclinations; (**b**) PolScope texture of the area indicated in part (**a**); (**c**) Corresponding velocity field of LCEO flows in the region marked as red rectangle in (**b**) caused by the AC electric field along the *x*-axis; (**d**) Corresponding velocity field of LCEO flows caused by the AC electric field acting along the *y*-axis in the same region**.** Adapted with permission from [26].

**Figure 12 micromachines-10-00045-f012:**
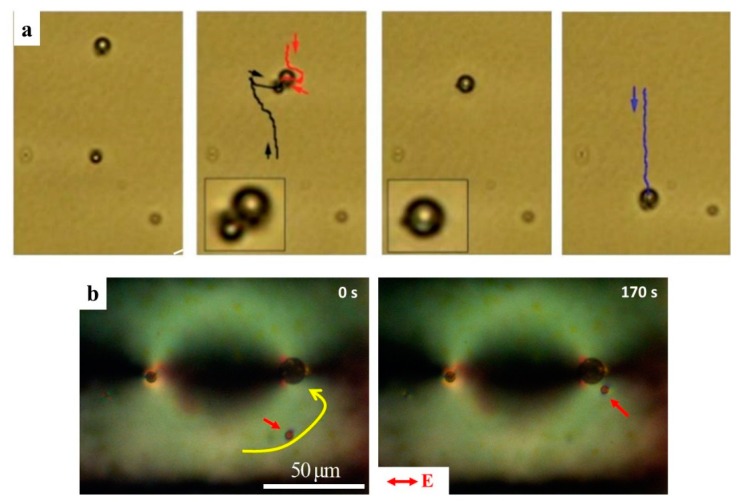
Liquid crystal-enabled electrokinetics (LCEK) applications in controlled microreactions. (**a**) Two water droplets stabilized by surfactants that impose perpendicular anchoring and loaded with specific reactants, are driven by LCEP in a uniformly aligned nematic chamber towards each other, until they coalesce and start a chemical microreaction [37]; (**b**) LCEO flows in a prepatterned nematic cell of the type shown in Figure 10a, with a (−½, ½) disclination pair, carry a surfactant-free water droplet (marked by a small arrow) towards a core of the ½ disclination where it coalesces with another droplet already trapped at the defect (the trajectory is shown by a curved arrow) [26]. Adapted with permission from [26,37].

**Figure 13 micromachines-10-00045-f013:**
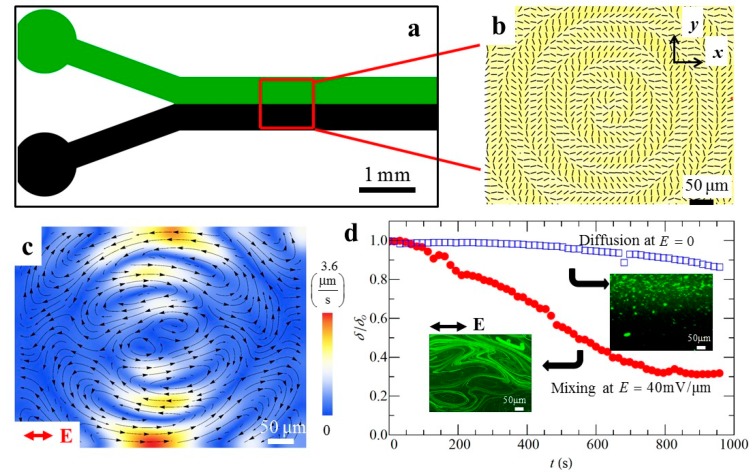
Micro-mixing in a Y-junction microfluidic device with photo-patterned director distortions demonstrated by Peng et al. [26]. (**a**) Y-junction with a photo-imprinted mixing pad (red square) that combines the LC with fluorescent particles (green) and pure LC (black); (**b**) PolScope texture of the mixing pad; (**c**) Velocity maps within the mixing pad caused by an AC electric field; (**d**) Comparison of mixing efficiencies of passive diffusion and LCEK. Adapted with permission from [26].

**Figure 14 micromachines-10-00045-f014:**
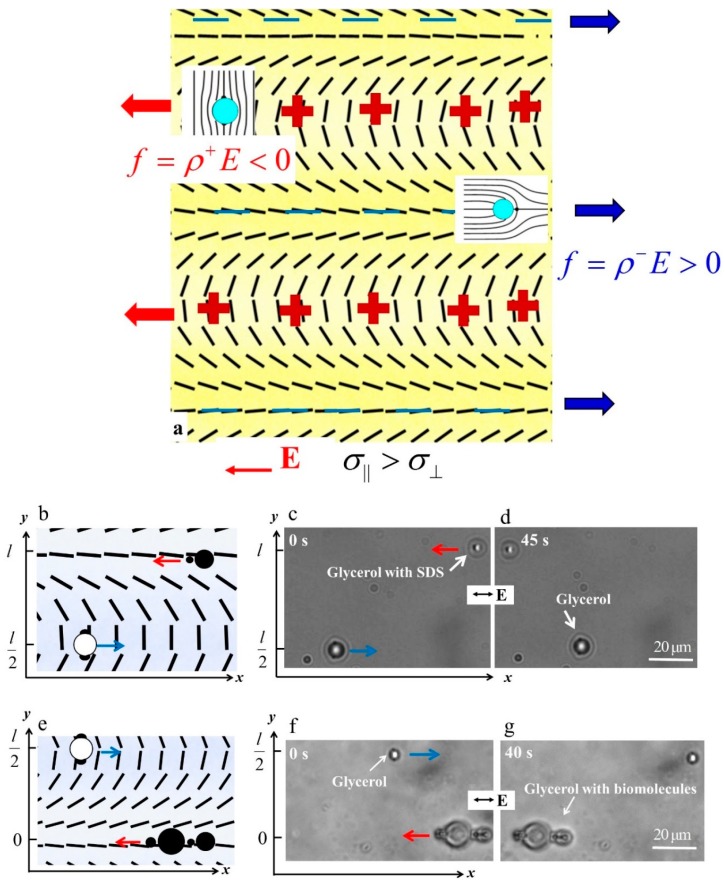
Sorting of fluid droplets with different surface properties by LCEO flows in patterned nematic cells by a method proposed by Peng et al. [70]. (**a**) Scheme of the director field, charge separation and preferential location of the colloids: spheres with normal anchoring accumulate in regions of splay, spheres with tangential anchoring accumulate in the regions of bend; (**b**–**d**) Sorting of droplets of pure glycerol (tangential anchoring) and glycerol with the surfactant SDS (perpendicular anchoring) by the applied AC electric field; The glycerol droplets with SDS in the splay region move to the left, while the pure glycerol droplets in the bend regions move to the right; (**e**–**g**) Sorting of droplets of pure glycerol and glycerol with biomolecules by the applied AC electric field; The biomolecules used are DLPC which are found in bio-membranes. Adapted with permission from [70].

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
