# Peer review of "Liquid Crystals-Enabled AC Electrokinetics"

_micromachines, 2019, doi:10.3390/mi10010045_

Round 1
Reviewer 1 Report
I believe the review should be published as it is with only minor corrections
1) line 222 correct "reinforce" by "reinforces"
2) lines 275/276 Please check "left" and "right". I think they should be the other way around: positive charges are guided towards the right (line 275) and negative charges are guided towards the left (line 276)
3) lines 448/448, please edit the sentence starting "Recently, Conklin...."
Author Response
We thank the Reviewer for evaluation and for useful comments.
1. We corrected misprint in "reinforce"
2. We checked "left" and "right": It was correctly stated, but we expanded the text in lines 273-280 to make it more clear. The field indeed moves positive charges to the right, but they are stopped by the left hand side of the sphere.
3. We edited the sentence in question
Reviewer 2 Report
This short review gives a good account of recent progress in the field of Liquid Crystal-enabled ElecroKinetics (LCEK), with a well-balanced mix of theoretical insights and experimental realisations. Interesting examples of electro-osmotic flow LCEO as well as electrophoresis (LCEP) (active particles) were given, and director field distortions by colloidal inclusions as well as by surface patterning were clearly discussed. The example of sorting particles and droplets is very illuminating, and also the possibility of defects as pumps. The underlying physics, in particular the key role played by the quadratic dependence on the applied electric field, is very well explained.
I therefore have no hesitation to recommend publication, although the manuscript would benefit from some more proof-reading and checking for English grammar here and there. For instance, I found:
line 71: show -> shows
line 211: Other -> Another
line 212: by the -> by
line 232: dynamically
line 266: in the presence
line 358: The director ...
line 367: a behavior -> the behavior
But surely a more critical eye will find quite a few more.
Author Response
We are thankful to the Reviewer for evaluation and useful remarks. We corrected all the typos mentioned. We also found a few places where English or technical writing could be improved and did polish the text.